Characterization and functional analysis of the Hydroxycinnamoyl-CoA: shikimate hydroxycinnamoyl transferase (HCT) gene family in poplar

Chao Nan 1 2 3
Qi Qi 2 4
Li Shuang 2
Ruan Brent 5
Jiang Xiangning 2 6
Gai Ying gaiying@bjfu.edu.cn 2 6
1 School of Life Science, Tsinghua University , Beijing , China
2 College of Biological Sciences and Biotechnology, Beijing Forestry University , Beijing , China
3 School of Biotechnology, Jiangsu University of Science and Technology , Zhenjiang , China
4 College of Horticulture, China Agricultural University , Beijing , China
5 Department of Agricultural and Biological Engineering, University of Illinois at Urbana Champaign , Urbana Champaign, IL , USA
6 National Engineering Laboratory for Tree Breeding, the Tree and Ornamental Plant Breeding and Biotechnology Laboratory of Chinese Forestry Administration , Beijing , China
Sun Genlou
Electronic publication date: 2021 Feb 25
Publication date: 2021
Volume: 9
Electronic Location ID: e10741
Received 2020 Feb 24; Accepted 2020 Dec 18
Copyright: ©2021 Chao et al.
Copyright year: 2021
Copyright holder: Chao et al.
License: This is an open access article distributed under the terms of the Creative Commons Attribution License, which permits unrestricted use, distribution, reproduction and adaptation in any medium and for any purpose provided that it is properly attributed. For attribution, the original author(s), title, publication source (PeerJ) and either DOI or URL of the article must be cited.
License URL: https://creativecommons.org/licenses/by/4.0/

Keywords: Hydroxycinnamoyl- CoA: shikimate hydroxycinnamoyl transferase, Enzymatic synthesis, Divergence, Gene family, Monolignol, Populus

Funding: The Beijing Higher Education Young Elite Teacher Project YETP0755 The National Natural Science Foundation of China NSF 31300498 This work was jointly supported by the Beijing Higher Education Young Elite Teacher Project [YETP0755 granted to Dr. YING GAI], the National Natural Science Foundation of China [NSF 31300498 to Ying Gai]. The funders had no role in study design, data collection and analysis, decision to publish, or preparation of the manuscript.

==============================
Hydroxycinnamoyl-CoA: shikimate hydroxycinnamoyl transferase (HCT) divides the mass flux to H, G and S units in monolignol biosynthesis and affects lignin content. Ten HCT homologs were identified in the Populus trichocarpa (Torr. & Gray) genome. Both genome duplication and tandem duplication resulted in the expansion of HCT orthologs in Populus. Comprehensive analysis including motif analysis, phylogenetic analysis, expression profiles and co-expression analysis revealed the divergence and putative function of these candidate PoptrHCTs. PoptrHCT1 and 2 were identified as likely involved in lignin biosynthesis. PoptrHCT9 and 10- are likely to be involved in plant development and the response to cold stress. Similar functional divergence was also identified in Populus tomentosa Carr. Enzymatic assay of PtoHCT1 showed that PtoHCT1 was able to synthesize caffeoyl shikimate using caffeoyl-CoA and shikimic acid as substrates.

Introduction

Primary walls and secondary walls protect plant cells and define the shapes of cells, tissues, organs and ultimately the whole plant body (Zhong, Cui & Ye, 2019). Lignin is an important component for secondary cell walls and is one of the most abundant components of biomass in plants (Boerjan, Ralph & Baucher, 2003; Tang & Tang, 2014). Therefore, lignin plays a vital role in plant physiology. Owing to the recalcitrant chemical nature and the complexity of lignin, lignin limits the conversion efficiency of lignocellulosic biomass to ethanol (Poovaiah et al., 2014; Vanholme et al., 2010). Modifying trees to have less lignin or more-degradable lignin along with normal growth, can reduce the high processing costs and carbon footprint of making paper, biofuels, and chemicals (Ralph, Lapierre & Boerjan, 2019; Tang & Tang, 2014; Wang et al., 2019; Xu & Li, 2016; Zhao, 2016).

The biosynthetic pathway for lignin has been studied extensively and the phenylpropane pathway which begins with phenylalanine, is responsible for monolignol biosynthesis. (Boerjan, Ralph & Baucher, 2003; Karkonen & Koutaniemi, 2010; Maeda, 2016; Ralph, Lapierre & Boerjan, 2019; Vanholme et al., 2010; Wang et al., 2019; Xu & Li, 2016). Monolignol is the general name for lignin building blocks. Our understanding of the monolignol biosynthetic pathway has continued to grow, and now 11 enzyme families and 24 metabolites are associated with it (Vanholme et al., 2019). Hydroxycinnamoyl- CoA: shikimate hydroxycinnamoyl transferase (HCT) is located at a key point in the monolignol biosynthetic pathway and is conserved across all land plants. In conjunction with C3H (p-coumarate 3-hydroxylase), HCT catalyzes two steps to direct the mass flux from the H monolignol to G and S monolignols (Fig. 1). HCT first catalyzes the coupling of p-coumaroyl-CoA with shikimate to produce p-coumaroyl shikimate (Hoffmann et al., 2004; Hoffmann et al., 2003). Caffeoyl shikimate is generated by C3H and is then transesterified by HCT to form caffeoyl-CoA. This reaction is probably reversible based on the reported in vitro activity (Lepelley et al., 2007; Wang et al., 2014). Caffeoyl shikimate esterase (CSE), a new member in monolignol biosynthesis pathway recently discovered in plants can hydrolyze caffeoyl shikimate to release caffeate (Ha et al., 2016; Saleme et al., 2017; Vanholme et al., 2013; Vargas et al., 2016). Although down-regulation of HCT expression improves forage digestibility and saccharification efficiency, it negatively affects plant growth resulting in shorter plants (Li et al., 2010; Shadle et al., 2007).

Figure 1 Schematic diagram of reaction catalyzed by HCT in monolignol biosynthesis pathway.

R=Shikimate. 4CL, 4-coumarate-CoA ligase; C3H, p- coumarate 3-hydroxylase; HCT, Hydroxycinnamoyl-CoA: shikimate hydroxycinnamoyl transferase. Compounds in red shadow are precursors for H units and in green shadow are for G and S units.

HCT (GO:0102660) belongs to the BAHD acyltransferase family and is able to utilize many non-native substrates. Some HCTs (also called HQT) can use quinate as a substrate in addition to shikimate (Eudes et al., 2016; Kim et al., 2013) for the biosynthesis of chlorogenic acid. As an acyl-CoA-dependent transferase, HCT is capable of acylating a wide variety of acceptors, with some exhibiting broad substrate flexibility (Chiang et al., 2018; Eudes et al., 2016). Crystal structures of HCTs from different plants have been determined for both the apo-form and complexed structure with diverse substrates allowing determination of active sites. For example, the apo-form and ternary complex with p-coumaroyl-CoA and shikimate of SbHCT from Sorghum bicolor (L.) Moench revealed the catalytic mechanism of HCT (Walker et al., 2013). Structures of AtHCT from Arabidopsis thaliana (L.), CbHCT from Coleus blumei Benth, CcHCT from Coffea canephora Pierre ex Froehn and SmHCT from Selaginella moellendorffii Hieron. have also been reported (Chiang et al., 2018; Lallemand et al., 2012; Levsh et al., 2016).

Similar to other key genes involved in monolignol biosynthesis, HCT is found as a gene family in many species in plant kingdom (Carocha et al., 2015; Ferreira et al., 2019; Ma et al., 2017; Raes et al., 2003; Zhang et al., 2018). The structural information and the proposed active sites of HCT, can help us to distinguish bona fide HCT utilizing shikimate as an acceptor and involved in monolignol biosynthesis in plants. In this study, we used genome-wide screening to identify 10 HCT homologs in Populus trichocarpa. Further motif and active site analysis showed the divergence of PoptrHCT s. Expression profiles and co-expression network analysis identified PoptrHCT1 and 2 as the lignin-related HCTs. Finally, we cloned and characterized the catalytic activity of PtoHCT1 from Populus tomentosa in vitro, which generated caffeoyl shikimate. PtoHCT1 could be used as the target gene for genetic modification to alter lignin content and composition.

Materials and Methods

Materials

Leaves of six—year-old Populus tomentosa 741 were collected from Hebei, China (Hu et al., 2019; Tian et al., 2013). Samples were immediately frozen in liquid nitrogen and then stored at −80 °C until use. Yu-Hu Ma approved sample collection at the study site. Yu-Hu Ma is the landowner of the study site, Shenzhou Famous and Excellent Seedling Breeding Base.

Genome-wide identification of HCT gene family members

To identify the HCT sequences in Populus, we first built a hidden Markov model (HMM) using reported HCT and HQT sequences. HMMsearch using Hmmer 3.0 software against the proteome data of Populus trichocarpa was performed based on the HMM model (Eddy, 2010). The cutoff for PoptrHCT homolog screening was an E-value (<E–100) of both the domain and full sequence and scores of full sequences (>400) (Table S1). The stable gene ID and symbols for HCTs reported in a previous study were also marked in Tables S1, S2. The sequences used for building the HMM model are shown in Table S2.

Distribution of HCT genes and HCT orthologs on Populus chromosomes

Ten candidate HCT and HCT orthologs were located on chromosomes in specific duplicated blocks which were determined based on the Populus genome and the WGDotplot in the PLAZA platform (Proost et al., 2009).

HCT sequence alignment and phylogenetic analysis

Alignment of PoptrHCTs and SbHCT, AtHCT were performed using DNAman 8.0 (Lynnon BioSoft) with default parameters. A phylogenetic tree was obtained using Mega 7.0 with the maximum-likelihood method (Kumar, Stecher & Tamura, 2016; Tamura et al., 2011). The phylogenetic tree was assessed by bootstrapping using 1000 bootstrap replicates and marked above nodes only if greater than 50. The JTT substitution model and G+I rates among sites model were selected as parameters for building the tree. The putative HCT sequences are listed in Table S2.

HCT expression profiles in P. trichocarpa and P. tomentosa

We obtained gene expression profiles for various tissues in P. trichocarpa using the GEO database with the accession number GSE30507. In addition, RNA-seq dataset GSE78953 including the transcriptome of various monolignol biosynthesis related mutants in P. trichocarpa, was used for co-expression analysis to explore the functions of the PoptrHCT orthologs. We also examined the expression profiles of PtoHCT orthologs in P. tomentosa in different seasons (Spring, Summer, Fall and Winter) and organs or tissues (roots, buds, phloem and xylem) using our microarray dataset (accession number: GSE56023 ) (Chao et al., 2014b). The corresponding PtoHCTs were identified using PoptrHCTs as queries by local blastn against the probe sequences database (Christiam et al., 2009) TBtools v0.6652 and Cytoscape 3.4 were used to visualize the HCT expression profile or co-expression network (Chen et al., 2020; Shannon et al., 2003) (Table S3).

Cloning and purification of recombinant HCT from P. tomentosa

Isolation of RNA and cDNA synthesis have been described in a previous study (Chao et al., 2014a). We cloned the homologous HCT1 from P. tomentosa based on the sequence information from P. trichocarpa (GenBank accession number: KT021003). Primer pair used for PCR amplification of PtoHCT1 is as follow: forward, 5′-CGATAAATAGAGCATTAGCACGGGG-3′; and reverse, 5′-ATAG CCTCGGCTCATTCTTT-3′. PCR products were purified and cloned into the pMD18-T vector (Takara Dalian), propagated in Escherichia coli DH5 α and inserts were confirmed by sequencing. PtoHCT1 was constructed with pET28a (Novagen) through a digestion-ligation way using restriction enzymes BamHI, HindIII and T4 ligation (Takara, Dalian). pET28a -PtoHCT1was then transformed into E.coli BL21(DE3). To induce expression, Isopropy-β-D-thiogalactoside (IPTG)was added to a final concentration of 0.8 mM and incubation was continued at 28 °C for four hours. Cells were collected by centrifugation at 4,000 g and 4 °C for 15min. The pellets were resuspended in lysis buffer (50 mM NaH2PO4, 300 mM NaCl with 10 mM imidazole, pH 8.0) and then disrupted by sonication. After centrifugation at 12,000 g and 4 °C for 30 min, the lysates were mixed with pretreated 1 ml Ni-NTA agarose (Qiagen Shanghai, China)) After washing using lysis buffer supplemented with 20 mM imidazole, the His-tagged PtoHCT1was eluted with 100 mM imidazole in lysis buffer.

Catalytic activity of recombinant PtoHCT1

Caffeoyl-CoA was chemically synthesized as reported (Chao et al., 2017). We determined the activity of recombinant PtoHCT1 by synthesis of caffeoyl shikimate using caffeoyl-CoA and shikimic acid as substrates. The reaction was performed according to Cesarino et al. (2013) and Luis et al. (2014). Briefly, total 40 µl standard reaction mix contained 100 mM Tris-HCl pH 7, 1mM DTT, 100 µM caffeoyl-CoA, 100 µM shikimic acid and 10 µg purified recombinant HCT protein. The reaction was initiated by adding the HCT proteins or the same amount of boiled protein as negative control. After Incubating at 30 °C for 30 min, the reaction was terminated by boiling the samples for 5 min. Flow for HPLC analysis was 0.1 mL/min in solvent A (acetonitrile) and solvent B (0.01% formic acid in water). The gradient was 0% A to 35% in B for 0 to 24 min, 35% A in B to 100% B for 24 to 27min, 100% B for 27 to 32min, 100% A to 100% B for 32 to 35 min, and 100% B for 35 to 55 min. The parameters used for MS analysis was sheath gas (nitrogen) flow rate, 40 arb; aux/sweep gas (nitrogen) flow rate, 10 arb; spray voltage,4.5 kV; capillary temperature, 320 °C. Optimized detailed parameters for dissociation of parent ions into product ions for each compound were provided in Table S4.

Structure modeling of PtoHCT1

The crystal structure of AtHCT (accession number 5KJT) (Levsh et al., 2016) was obtained from the Protein Data Bank to build a homolog model for PtoHCT (https://www.rcsb.org). Molecular docking was performed using CDOCKER assembled in Discovery Studio 4.5. Visualization of the active sites and 3-D structures were generated by Discovery Studio 4.5.

Results

Genome-wide identification and distribution of HCT orthologs in Populus

Ten PoptrHCT homologs were found based on HMMsearch against the Populus genome (Table S1). These HCT candidate genes are located on six different chromosomes. Among the 10 PoptrHCT orthologs, PoptrHCT3, 4 and 5 were located on chromosome V, and PoptrHCT7, 8, 9, and 10, were on chromosome XVIII, representing two clusters respectively (Fig. 2A). Tandem duplication is likely to be responsible for the formation of HCT homolog clusters. Ks (substitution per synonymous site) value distributions can be used for revealing whole genome duplication (WGD) events (Jiao et al., 2011; Tang et al., 2010). PoptrHCT1 and PoptrHCT2 formed a homolog duplicate pair with Ks value 0.2174 and were located at corresponding homologous duplicated blocks, as the result of whole genome duplication. The organization of the PoptrHCT orthologs indicates that both genome duplication and tandem duplication played roles in the formation of the HCT family.

Figure 2 PoptrHCT orthologs organization and phylogenetic analysis.

(A) Organization of HCT orthologs on Populus chromosomes. Regions that are assumed to correspond to homologous genome blocks are shaded gray and connected by lines. The position of genes is indicated with an arrowhead. (B) Phylogenetic analysis of HCT homologs from Populus trichocarpa and other plant species. The PoptrHCT1 and 2 were marked with full black triangle. Two groups for HCT orthologs were shown and HCTs in Group I are likely to transfer hydroxycinnamates to shikimate and have been implicated in monolignol biosynthesis. The scale bar indicates 0.5 amino acid substitutions per site in given length. The accession numbers of sequences used are as followed: Arabidopsis thialiana AtHCT (AT5G48930), AtHCTlike (AT4G29250); Amborella trichopoda AtrHCT1 (ATR_00137G00320), AtrHCT2 (ATR_00727G00010); Cynara cardunculus CcaHCT (DQ104740), CcaHQT(ABK79690); Lycopersicon esculentum LeHQT (AJ582652); Larix kaempferi LkaHCT (AHA44839); Nicotiana tabacum NtaHCT (Q8GSM7), NtaHQT (CAE46932); Picea lauca PglHCT (CZO01061061); Populus trichocarpa PoptrHCT1 (PT01G04290), PoptrHCT2 (PT03G18390), PoptrHCT3 (PT05G02800), PoptrHCT4 (PT05G02810), PoptrHCT5 (PT05G02840), PoptrHCT6 (PT18G03270), PoptrHCT7 (PT18G10470), PoptrHCT8 (PT18G10480), PoptrHCT9 (PT18G10540), PoptrHCT10 (PT18G10550); Physcomitrella patens PpHCT1 (PP00022G00830); Panicum virgatum PviHCT1a (JX845714), PviHCT2 (KC696573 ); Sorghum bicolor SbHCT (XP_002452435.1).

Alignment and phylogenetic analysis of HCT orthologs

Putative protein sequences of PoptrHCT orthologs and crystal structures of two shikimate-specific HCTs (AtHCT and SbHCT) and LeHQT (Lycopersicon esculentum Mill.) were aligned. Characteristic of the BADH superfamily, two motifs HXXXD(G) and DFGWG were conserved in AtHCT , SbHCT and all PoptrHCT orthologs (except PoptrHCT6) (Fig. 3) (D’Auria, 2006). Based on previous studies of the structure of HCTs including site-directed mutagenesis, molecular docking and crystallographic analyses we summarized the active sites of HCTs (Table 1) and marked these active sites in Fig. 3. Active sites for the carbonyl group of the p- coumaroyl moiety binding and the catalysis related sites of LeHQT correspond with HCTs (red full circles) while divergence is obvious in terms of active sites for shikimate binding (red full stars). PoptrHCT1 and PoptrHCT2 showed conservation at these active sites and kept correspondence with AtHCT and SbHCT, while PoptrHCT3-10 showed poor conservation at these key sites. Thus while the ten candidate PoptrHCTs mostly belong to the BADH superfamily, only PoptrHCT1 and PoptrHCT2 appear to be associated with monolignol biosynthesis. Phylogenetic analysis showed PoptrHCT1 and PoptrHCT2 grouped with Group I HCTs, which transfer hydroxycinnamates to shikimate and have been implicated in monolignol biosynthesis, strongly suggesting this role for PoptrHCT1 and 2 as well (Fig. 2B). Other PoptrHCTs (3–10) clustered with HQTs and other HCT-like as Group II and especially, PoptrHCT6 without DFGWG seem unlikely to be shikimate-specific transferases involved in monolignol biosynthesis. Thus these Group II. PoptrHCTs might have different catalytic activity (e.g., utilize acceptors other than shikimic acid) and are likely to play different roles in plants.

Figure 3 Alignment of PoptrHCT and PoptrHCT orthologs compared to shikimate-specific HCTs from Arabidopsis and sorghum and HQT from tomato.

Red full stars indicate shikimate binding sites, red full circles indicate carbonyl group of p-coumaroyl moiety binding sites, purple full triangle indicate carbonyl group of shikimate moiety binding sites and the blue full circles indicate sites involved in catalysis. Accessions are as in Fig. 2. Detailed references are also available in Table 1.

Table 1 Summary of active sites of HCTs.

Position	Amino acid	Annotation	Reference	
31	Val	Shikimate binding	Walker et al. (2013)	
32	Pro	Shikimate binding	Walker et al. (2013)	
298	Ala	Shikimate binding	Walker et al. (2013)	
318	Ile	Shikimate binding	Walker et al. (2013)	
376	Phe	Shikimate binding	Walker et al. (2013)	
414	Leu	Shikimate binding	Walker et al. (2013), Lallemand et al. (2012)	
418	Leu	Shikimate binding	Walker et al. (2013)	
38	Ser	carbonyl group of p- coumaroyl moiety	Walker et al. (2013); Eudes et al. (2016)	
40	Tyr	carbonyl group of p- coumaroyl moiety	Walker et al. (2013), Lallemand et al. (2012), Eudes et al. (2016)	
384	Trp	carbonyl group of p- coumaroyl moiety	Walker et al. (2013)	
163	His	carbonyl group of shikimate moiety and Catalysis	Walker et al. (2013), Lallemand et al. (2012), Eudes et al. (2016)	
369	Arg	carbonyl group of shikimate moiety	Walker et al. (2013)	
382	Thr	carbonyl group of shikimate moiety	Walker et al. (2013), Eudes et al. (2016)	
36	Thr	Catalysis	Walker et al. (2013)	
Notes.

All positions correspond to SbHCT.

Expression analysis of HCT homolog genes

Based on the microarray analysis of seven different tissues and organs in P. trichocarpa, PoptrHCT1 and 2 showed expression preference in developing xylem (DX) and mature xylem (MX). PoptrHCT1 showed high expression levels in all detected tissues and organs. PoptrHCT9 and 10 showed expression preference in developing tissues including developing phloem, developing xylem, cambium (C) and shoots and leaf primordium (SLp) (Fig. 4A). Co-expression network analysis shows that PoptrHCT1 and 2 have significant correlations with genes involved in lignin biosynthesis (Fig. 4B). Monolignol biosynthesis related transcription factors also showed co-expression with PoptrHCT1 and 2. The similar expression patterns were also found in P. tomentosa Carr. (Pto). PtoHCT1 has high expression levels in almost all tissues and organs year- round while PtoHCT9 and 10 showed preference in buds and phloem especially in winter, which indicates that these two genes could be involved in development of dormancy and response to cold stress. The differential expression of the PoptrHCT orthologs further supports that HCT1 plays a major role in monolignol biosynthesis.

Figure 4 Expression profile and co-expression network of HCT orthologs in poplar.

(A). Expression profile of HCT orthologs in Populus. Tissues or specific parts of plants are indicated with the respective abbreviations: WS, whole stems; BM, Bark and mature phloem; C, cambium; DP, developing phloem; DX, developing xylem; ML, mature leaf; SLp, shoot and leaf primordium. (B) Expression profile of HCT orthologs in P. tomentosa. (C) Co-expression network of PoptrHCT orthologs with identified genes involved in lignin biosynthesis. The support information is available in Table S3. Only nodes with Pearson correlation coefficients ¿0.9 were shown and considered as close co-expression.

Catalytic activity and structure comparison of PtoHCT1

PtoHCT1 protein expressed in E. coli was purified for enzymatic assays. We monitored PtoHCT1 reactions using HPLC-MS and found that PtoHCT1can utilize caffeoyl-CoA and shikimic acid (Fig. 5). After the initiation of the reaction by adding PtoHCT1, the accumulation of caffeoyl shikimate and decrease of caffeoyl-CoA and shikimic acid were observed within 2 min. We built a homology model for PtoHCT1 to explore the structure of PtoHCT1 using the crystal structure of AtHCT (accession number 5KJT) as template. The main-chain root-mean-square deviation (RMSD) is 0.224 Å indicating the high structure similarity of PtoHCT1 and AtHCT (Fig. 6A). According to the summarized active sites (Table 1), we found these conserved sites around the catalytic cleft, and then we successfully docked PtoHCT1 with the substrate caffeoyl-CoA, which further provided the structural evidence for PtoHCT1 catalyzing caffeoyl-CoA (Fig. 6B).

Figure 5 PtoHCT1 catalyzes enzymatic synthesis of caffeoyl shikimate.

LC separation of reactions with MS detection (selected ion signals) (A) at initiation of the reaction (B) after 80s or (C) after 120s.

Figure 6 The structure of PtoHCT1 and docking with caffeoyl-CoA.

(A) Structure alignment of AtHCT (green) and PtoHCT (purple). (B) PtoHCT docked with caffeoyl-CoA. Blue ligand is caffeoyl-CoA and active sites are labeled.

Discussion

HCT regulates the flux at a key point in monolignol biosynthesis and has been studied in many plants (Hoffmann et al., 2004; Hoffmann et al., 2003; Shadle et al., 2007; Sun, Yang & Tzen, 2018; Wagner et al., 2007). HCT also exists as a gene family in the land plant kingdom similar to other key genes involved in monolignol biosynthesis. We focused on the HCT gene family in this study and provided a systematic analysis of the HCT genes in poplar, and identified two lignin-related HCTs (HCT1 and HCT2), which exist as a homolog pair located at a duplication block on chromosome I and III respectively (Fig. 2A). Ks analysis indicated that the HCT gene pair (PoptrHCT1 and PoptrHCT2) resulted from a recent genome duplication (Ks value 0.2174). Two HCT homolog clusters resulting from tandem duplication were also identified. Both genome duplication and tandem duplication provide raw genetic material for neo-function as a driving force in plant evolution and are responsible for the expansion of HCT orthologs (Zhang, 2003).

Based on our systematic analysis, PoptrHCT1 and 2 are involved in monolignol biosynthesis. Both PoptrHCT1 and 2 showed expression preference in xylem and co-expression with other monolignol related genes. HCT1 (PoptrHCT1 and PtoHCT1) had high expression levels in different tissues. PtoHCT1 showed catalytic activity for caffeoyl-CoA and shikimic acid. These results further validate HCT1 as the dominant HCT in monolignol biosynthesis, while HCT9 and 10 (PoptrHCT9, 10 and PtoHCT9, 10) showed expression preference in developing tissues. PtoHCT9 and 10 were active in winter, which suggested a function in plant development and response to cold stress. PtoHCT1 utilizes caffeoyl-CoA and shikimic acid to generate caffeoyl shikimate which can be used as substrate for CSE, a new annotated enzyme involved in monolignol biosynthesis (Ha et al., 2016; Saleme et al., 2017; Vanholme et al., 2013).

Conclusion

In summary, we identified ten HCT homologs and proposed the important roles of both genome duplication and tandem duplication in the expansion of HCT orthologs in Populous. Two HCTs likely involved in monolignol biosynthesis in Populus were identified based on phylogenetic analysis and expression profile analysis. Enzymatic assay of PtoHCT1 showed that PtoHCT1 was able to synthetize caffeoyl shikimate using caffeoyl-CoA and shikimic acid as substrates. In addition, other PoptrHCT orthologs showed divergence in reported active sites and different expression pattern. HCT9 and 10 (PoptrHCT9, 10 and PtoHCT9, 10) showed preferential expression in developing tissues and were active in winter. Further studies should help to reveal the functions of the other HCT orthologs.

Supplemental Information

Supplemental Information 1 Genome-wide screening PoptrHCT genes based on HMM models

Click here for additional data file.

Supplemental Information 2 Putative HCTs used for phylogenetic analysis

Click here for additional data file.

Supplemental Information 3 Expression profiles and co-expression for HCT homologs

Click here for additional data file.

Supplemental Information 4 Conditions used for HPLC-MS to identify caffeic acid, shikimate and caffeoyl shikimate

Click here for additional data file.

Supplemental Information 5 MIAME.

Click here for additional data file.

Supplemental Information 6 MIAME for GEO

Click here for additional data file.

Supplemental Information 7 Sequences used for alignment analysis

Click here for additional data file.

Additional Information and Declarations

Competing Interests

Author Contributions

Field Study Permissions

Data Availability

The authors declare there are no competing interests.

Nan Chao performed the experiments, analyzed the data, prepared figures and/or tables, authored or reviewed drafts of the paper, and approved the final draft.

Qi Qi performed the experiments, analyzed the data, prepared figures and/or tables, and approved the final draft.

Shuang Li performed the experiments, prepared figures and/or tables, and approved the final draft.

Brent Ruan performed the experiments, authored or reviewed drafts of the paper, and approved the final draft.

Xiangning Jiang and Ying Gai conceived and designed the experiments, authored or reviewed drafts of the paper, and approved the final draft.

The following information was supplied relating to field study approvals (i.e., approving body and any reference numbers):

Yu-Hu Ma approved sample collection at the study site. Yu-Hu Ma is the landowner of the study site, Shenzhou Famous and Excellent Seedling Breeding Base.

The following information was supplied regarding data availability:

The sequence is available at NCBI GEO: GSE56023. Additional data are available in the Supplemental Files.

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
