# Peer review of "Characterization and functional analysis of the Hydroxycinnamoyl-CoA: shikimate hydroxycinnamoyl transferase (HCT) gene family in poplar"

_PeerJ, doi:10.7717/peerj.10741_

## Round 0.1 · original submission · Major Revisions

Two reviewers have made comprehensive comments on your manuscript. Please revise your manuscript accordingly.

Reviewer 1 ·

Basic reporting

Overall the English in the paper is good, and it was not difficult to read from that perspective, although I saw some grammar and word choice issues with suggested corrections that are noted under general comments.

Article structure is appropriate, although if the journal allows, I'd suggest combining results and discussion section as I think this would be a more effective presentation.

Most figure legend lack sufficient detail (see general comments)

Experimental design

There are two main structural problems I have with this manuscript. It may not be so much an experimental design issue, as much as how the results are presented in the manuscript.

I think it ends up being confusing to refer to the 10 genes as an “HCT gene family” if the authors are defining “HCT” as hydroxycinnamoyl-CoA:SHIKIMATE hydroxycinnamoyl transferases. I think their data suggest that PoptrHCT1 and HCT2 are likely transferases capable of using shikimate as acceptor substrate (and involved in biosynthesis of monolignols), but if I’m interpreting their data correctly it seems likely several/many/most/all of the others are hydroxycinnamoyl-CoA hydroxycinnamoyl transferases that could have different acceptor substrate specificities (e.g. quinate, malate, hydroxyphenylactate). If I am correct here, the title of the paper ends up being inaccurate. There is admittedly some sloppiness in the field in the use of “HCT”, since the most famous, the shikimate transferases implicated in monolignol biosynthesis, are usually just referred to as HCTs, but the term is also used generically for hydroxycinnamoyl-CoA transferases regardless of acceptor substrate. Thus, I think authors need to be very specific in their writing about these enzymes, especially if they might be talking about both enzymes with a specific substrate and implicated function (i.e. the shikimate transferases involved in conversion of p-coumaric acid to caffeic acid) and the class of enzymes in general (i.e. hydroxycinnamoyl-CoA transferases regardless of what acceptor substrate they use).

I am also somewhat confused by the Populus tomentosa data suddenly showing up around line 150. Maybe it is my lack of familiarity with poplar. What is the relationship between Populus trichocarpa and Populus tomentosa and between the HCT genes? It’s sort of strange for the Pto data to suddenly show up in this section without prior mention of it. Has a similar analysis of genes from P. tomentosa already been done, and if so, how does the work here on P. trichocarpa add to knowledge? From context, it seems like there’s a 1 to 1 correspondence between these two species. I think this needs to be better addressed in some way.

The methodology for the biochemical experiment is insufficient. More detail must be provided. See general comments

Validity of the findings

This manuscript details a study using genomics to identify a family of hydroxycinnamoyl-CoA hydroxycinnamoyl transferases from Populus trichocarpa and then use expression analysis to identify those likely involved in lignin biosynthesis. I believe the underlying work is useful and should be publishable, but I think the manuscript needs revision prior to publication. In some cases more detail is needed, and being more focused on some details may help the authors make their case.

Additional comments

Below is the initial review I carried out. I copied and pasted as appropriate into the Reporting, Design, and Validity sections above as appropriate from this review, so some comments below will be redundant. Please note, there are many specific comments below not covered in the other sections.

This manuscript details a study using genomics to identify a family of hydroxycinnamoyl-CoA hydroxycinnamoyl transferases from Populus trichocarpa and then use expression analysis to identify those likely involved in lignin biosynthesis. I believe the underlying work is useful and should be publishable, but I think the manuscript needs revision prior to publication. In some cases more detail is needed, and being more focused on some details may help the authors make their case.

Importantly, I think it ends up being confusing to refer to the 10 genes as an “HCT gene family” if the authors are defining “HCT” as hydroxycinnamoyl-CoA:SHIKIMATE hydroxycinnamoyl transferases. I think their data suggest that PoptrHCT1 and HCT2 are likely transferases capable of using shikimate as acceptor substrate (and involved in biosynthesis of monolignols), but if I’m interpreting their data correctly it seems likely several/many/most/all of the others are hydroxycinnamoyl-CoA hydroxycinnamoyl transferases that could have different acceptor substrate specificities (e.g. quinate, malate, hydroxyphenylactate). If I am correct here, the title of the paper ends up being inaccurate. There is admittedly some sloppiness in the field in the use of “HCT”, since the most famous, the shikimate transferases implicated in monolignol biosynthesis, are usually just referred to as HCTs, but the term is also used generically for hydroxycinnamoyl-CoA transferases regardless of acceptor substrate. Thus, I think authors need to be very specific in their writing about these enzymes, especially if they might be talking about both enzymes with a specific substrate and implicated function (i.e. the shikimate transferases involved in conversion of p-coumaric acid to caffeic acid) and the class of enzymes in general (i.e. hydroxycinnamoyl-CoA transferases regardless of what acceptor substrate they use).

I am also somewhat confused by the Populus tomentosa data suddenly showing up around line 150. Maybe it is my lack of familiarity with poplar. What is the relationship between Populus trichocarpa and Populus tomentosa and between the HCT genes? It’s sort of strange for the Pto data to suddenly show up in this section without prior mention of it. Has a similar analysis of genes from P. tomentosa already been done, and if so, how does the work here on P. trichocarpa add to knowledge? From context, it seems like there’s a 1 to 1 correspondence between these two species. I think this needs to be better addressed in some way.

If allowed by PeerJ, I would suggest combining the Results and Discussion section.

Most of the figure legends lack sufficient detail to be understandable on their own.

Overall the English in the paper is good, and it was not difficult to read from that perspective, although I saw some grammar and word choice issues with suggesrted corrections that are noted below.

Some specific comments:

Line 21: for the sentence starting with “Ten HCT homologs…” I would split it into two sentences and be more conservative. “….. genome. Both genome duplication and tandem duplication are likely responsible for the expansion…”

Line 36: insert “a” before “vital”

Line 38: replace “with” with “to have”

Lines 44-45: the pathway hasn’t been improved upon, our understand has. Possible rewording “Our understanding of the monolignol biosynthetic pathway has continued to grow, and now 11 enzyme families and 24 metabolites are associated with it.”

Line 47: replace “Combining” with “In conjunction”.

Line 52: replace “revisable” with “reversible”, replace “enzymatic activity” with “in vitro activity”.

Line 55: Perhaps more context could be given for the energy-loss bypass mentioned here. Why does this matter? What’s its relevance here?

Line 57: Perhaps mention that the effect on plant growth is negative.

Line 73: The sentence is awkward. I’d suggest “In this study, we used genome-wide screening to identify 10 HCT homologs...”

Line 97: The phylogenetic tree shown in Figure 2B does not completely correspond with the list of HCTs in Supplementary Table 2. Some things listed in the supplement are not in the tree, and some things in the tree are not in the supplement.

Line 112: replace “addition” with “agent”, replace “with” with “to”

Line 114: insert “supplemented with” after “buffer”.

Line 115: replace “were” with “was”, delete “finally”

Lines 116-120: Authors need to provide more detail here. Some aspects of the experiment here are clearly different than the cited work by Luis et al. For example, detection method used here seems to be MS, but that’s not clear, while Luis et al. used UV-Vis detection (although they were not as specific in their methodology either). It would be helpful for a reader to know more about the reaction conditions used (how much substrate, how much enzyme) here without having to go back to another paper.

Line 128: replace “screened” with “identified”

Line 130: addition of commas would help make this more clear, “… poptrHCT3, 4, and 5, distributed on chromosome V, and PoptrHCT7, 8, and 9, distributed on chromosome XVIII,”

Line 132: For readers not familiar with analysis of genome structure (like me), perhaps some explanation on how to interpret the Ks score would be helpful. See also line 184.

Line 148 and 149: use of the word “genuine” here seems inappropriate. Use more specific language for what you mean. By “genuine” do you mean “uses shikimic acid as an acceptor and has a role in monolignol biosynthesis”? If so, use the more specific language. I think more details about the phylogenetic tree could also be given. Looking at the figure, there’s no mention in the text about “Group I” and “Group II”. What are those groups? On the figure, there’s a scale bar indicating 0.5. This is outside of my area…. What does that value represent? Also, I think some, if not all, of the information from Supplementary Table 2 could be included in the figure legend.

Line 150-160: Mentioned above, this section is somewhat confusing to me. Maybe it is my lack of familiarity with poplar. What is the relationship between Populus trichocarpa and Populus tomentosa and between the HCT genes? It’s sort of strange for the Pto data to suddenly show up in this section without prior mention of it. Has a similar analysis of genes from P. tomentosa already been done, and if so, how does the work here on P. trichocarpa add to knowledge? Additionally, for Figure 4A, what does the numeric scale represent? I don’t find Figure 4B particularly helpful… is this a standard way to show these relationships? It seems like there might be some better way to show that HCT1 and 2 have significant coexpression with monolignol biosynthetic genes, while the others don’t.

Line 163: I would suggest rewriting the sentence as “Recombinant PtoHCT1 protein was expressed in E. coli and purified to examine enzymatic activity.”

Line 165 and 173: I think it is confusing to refer to transfer of caffeic acid from caffeoyl-CoA to shikimic acid as the “reverse reaction”. These BAHD hydroxycinnamoyl-CoA transferases generally can use a variety of hydroxycinnamoyl-CoAs in vitro, including p-coumaroyl-, caffeoyl-, and feruloyl-CoA. I believe transfer of the hydroxycinnamate from the CoA derivative to acceptor (in this case shikimic acid) is viewed as the forward reaction. Reverse reaction is, I think, generally considered reaction of a hydroxycinnamoyl deriviative (here hydroxycinnamoyl-shikimate) with CoA to reform hydroxycinnamoyl-CoA and release free acceptor (here shikimate).

Lines 178-181: I don’t really understand what point the authors are trying to make.

Line 182: delete “finally”.

Line 189-196: do the authors believe that other members of this family, e.g. HCT3-10, might have different acceptor substrate preferences (i.e., something other than shikimic acid)? That might be worth mentioning here.

Line 197: does not need to be included in the paper, but depending on what the authors intend with CSE, chlorogenic acid might be a suitable substrate for some studies of the enzyme, plus it is readily available and inexpensive.

Line 208: “Advice” not “advices”

Figure legends in general need more detail. Figures should be overall understandable without having to refer to text or supplementary material. More specifically:

Figure 1: Please define all enzyme abbreviations in the legend. Should also include 4-coumarate CoA ligase (4CL). Caffeic acid on the right side is missing its hydroxyl.

Figure 2: Legend in 2B could include all the genes shown on the tree. Readers shouldn’t be sent to the supplement to know what genes and species are analyzed in the tree. As mentioned above, please define what the scale bar represents.

Figure 4: As mentioned above, please define what the heat map scale corresponds to. I don’t find 4B very helpful. Is there a better why to visualize the co-expression data. This is somewhat outside my field, but might a “heat map” type of visualization with the Pearson correlations as the value be effective? It could be a matrix of all the HCTs versus several genes implicated in monolignol biosynthesis. HCT1 (and 2) would be hot, HCT 3-10 would be cold (I presume). This is just an idea, but seems to get at the core of what the authors are trying to convey.

Figure 5: Although overall one can see shikimate disappearing and caffeoyl-shikimate appearing, the authors provide too little detail in the methods and in this figure legend to understand what a reader is looking at. I assume this is MS (following LC separation?), but what signal are we looking at in the chromatograms (total negative ion, total positive ion, selected ions?) Are the insets what we should expect? I guess I would not automatically know what to expect those insets to look like (I’ve seen MS detection for these sorts of compounds set up to give a major ion of +1 or -1 from the expected molecular weight, depending on polarity).

Reviewer 2 ·

Basic reporting

.

Experimental design

.

Validity of the findings

.

Additional comments

See attachment.

Specific comments
line 25: please explain what functions of lignin are being considered.
line 28: systematic investigations: in what way? explain.
line45: "improving the pathway" meaning not clear.
line 52: do you mean reversible?
line 59: when using words line a variety, or many. be more precise.
line 72: distinguish more accurately.... please expain.
line 77:further CSE analysis, please explain.
line 86: more details needed on building and validating
line 140: referred HCTs, meaning not clear.
line 143 p- should be in italics in the text and in the figures.
line 173: what does it all mean? What has been learned from the study?
line 179and following. meaning not clear, confusing sentence. Please revise.
figure 1. p should be in italics.
figure 1. Which metabolites give rise to H units in lignin (from p-coumarate) and C units, G units and S units
figure 1.p-coumaroyl shikimate needs to be corrected.
figure 4. much more can be said about the mybs an other lignin gene associations.
figure 4> "homology genes" should be "homologs"

Annotated reviews are not available for download in order to protect the identity of reviewers who chose to remain anonymous.

---

## Round 0.2 · Major Revisions

This manuscript requires major revision.

Reviewer 1 ·

Basic reporting

Basic structure of the manuscript is fine. Sufficient background is provided, and it is self contained.

I think the use of the term "HCT" is still problematic. I think better distinction needs to be made between HCT as the shikimate-utilizing transferase implicated in monolignol biosynthesis, and other BAHD hydroxycinnamoyl-CoA transferases that (probably) use other acceptor substrates and have other functions. I personally think saying "HCT-like" or "Bona fide HCT" does not adequately describe to a reader the distinction here. It may end up being a matter of adding additional text to make sure it is clear. I have attempted to do so in the attached copy which has "track changes" correction suggestions and comments

I think it might also be helpful to use the more specific term "orthologs" instead of "homologs" where appropriate (i.e., to refer to the homologs/gene family within the species).

Despite many editing suggestions in the previous version by both myself and the other reviewer, I still found many corrections for English and grammar were needed. Again, see the attached document.

Experimental design

Overall OK. More detail is needed for how the expression construct was made and protein purified. I did not notice this was missing in the last version, but a bit more detail should be provided.

Validity of the findings

OK. I made a few comments where I believe certainty of conclusions was stated too strongly and needed to be softened.

Additional comments

The gist of the paper is 10 HCT orthologs/homologs were identified. Based on phylogeny and structure, two (1 and 2) are likely shikimate-specific for acceptor substrate and associated with monolignol biosynthesis. Two (9 and 10) MAY be involved in dormancy and cold tolerance. The others likely have some other function.

Annotated reviews are not available for download in order to protect the identity of reviewers who chose to remain anonymous.

---

## Round 0.3 · Minor Revisions

Here are further comments from reviewer 3. Please revise your manuscript accordingly.

Reviewer 3 ·

Basic reporting

The work by Chao et al. describes phylogenetic analysis of the HCT gene family in poplar. Additionally, the authors performed enzyme assays to implicate PtoHCT1 in caffeoyl shikimate biosynthesis suggesting a role in lignin biosynthesis. The HCT gene family is one of the most studied family in poplar, as such most of the conclusions drawn by the authors are confirm what previous studies have reported. This includes characterization of whether family members evolved from the whole genome duplication event or from tandem duplication (Zhang et al. 2018). Additionally, functional specialization into lignin biosynthesis or stress response has also been reported (Zhang et al. 2018).

Specific comments

1. Authors should consider reconciling their naming nomenclature with existing literature. It appears that they number HCT1-10 by increasing position on chromosomes as well as increasing chromosome number. Previous studies do not follow this pattern and it can be confusing to readers.
2. Authors should also use the full gene model that is available in phytozome, e.g PT01G04290 should be Potri.001G042900 as found in v3.1 of the P. trichocarpa reference genome.
3. Previous studies report 9 members of the HCT gene family in poplar. Here, the authors report 10 including PT18G03270 (Potri.018G032700) annotated as a ‘HXXXD-type acyl-transferase-like protein’ in the poplar reference genome. The author should justify and explain its inclusion and relationship to other well established members of the family.

Reference

Zhang, Jin, et al. "Genome‐wide association studies and expression‐based quantitative trait loci analyses reveal roles of HCT 2 in caffeoylquinic acid biosynthesis and its regulation by defense‐responsive transcription factors in Populus." New Phytologist 220.2 (2018): 502-516.

Experimental design

Experimental design was appropriate for the study.

Validity of the findings

Findings confirm what has been previously reported about the HCT gene family with the exception of the number of family members as noted in the review comments.

---

## Round 0.4 · Minor Revisions

Please address these changes and resubmit.

Reviewer 1 ·

Basic reporting

I believe the authors have addressed most issues in the manuscript, and it is acceptable for publication pending minor corrections within the manuscript. Many of these I will not detail, but should be able to be corrected by a copy editor.

A few specific issues:

Line 30, “components” instead of “component”.
Line 45, p in p-coumaroyl should be italicized
Line 55, what is the “t” for?
Line 65, I’d suggest again clarifying what you mean by saying “bona fide HCT utilizing shikimate as an acceptor and involved in monolignol biosynthesis”.
Line 69, there seems to be an extra “.”
Line 81, “… in a previous study are also shown in Supplementary….”
Line 105, “… in a previous study…”
Line 164, instead of “is far from…”, I would again suggest “seem unlikely to be shikimate-specific transferases involved in monolignol biosynthesis”
Line 206, use “utilizes” instead of “catalyzes”

Experimental design

OK

Validity of the findings

OK

Additional comments

See above

---

## Round 0.5 · Minor Revisions

The manuscript reads well and data sources appear well developed and pointed to. As the manuscript characterizes a gene family to functional expression, a refined annotation of the family members is warranted; preferable and perhaps easily done using gene ontology terms. As the main focus is primarily on two of the family I may forgo this additional step, but it would be an enhancement and logical step for the manuscript.

Journal manuscripts are often scanned by text-mining software that locates and extracts core data elements, like gene function. Adding standard ontology terms, such as the Gene Ontology (GO, geneontology.org) or others from the OBO foundry (obofoundry.org) can enhance the recognition of your contribution and description. This will also make human curation of literature easier and more accurate. None of this was visible.

The only connection to sequence data is found in Figure 3 which is difficult for the reader to work with. It is requested that a supplemental file with the sequence data be provided in FASTA format, or better yet, in the DNA context for the transcripts. Navigating the descriptions pointed to in resources does not provide a readily available form of the sequences described.

The manuscript requires minor revisions, but would like the authors to consider adding the gene annotations in some form.

---

## Round 0.6 · accepted · Accept

I am pleased to inform you that your revised manuscript is accepted.